# Distress of Educators Teaching Nursing Students with Potential Learning Disabilities: A Qualitative Analysis

**DOI:** 10.3390/healthcare11040615

**Published:** 2023-02-18

**Authors:** Mari Murakami, Hiromi Kawasaki, Zhengai Cui, Hiroko Kokusho, Masayuki Kakehashi

**Affiliations:** Graduate School of Biomedical and Health Sciences, Hiroshima University, 1-2-3 Kasumi, Minami-ku, Hiroshima 734-8553, Japan

**Keywords:** nursing university students, learning disabilities, educational support, online, focus group interviews, qualitative research

## Abstract

Clinical training at Japanese nursing universities has an increasing need for individualized learning support for students with potential learning disabilities. Despite a high interest in student support, educators’ difficulties are neglected. This study clarified the difficulties encountered by practical training instructors in delivering clinical training to nursing students with potential learning disabilities. In this descriptive, qualitative study, online focus group interviews were conducted. Participants were nine Japanese nursing university graduates with over five years of clinical education experience. A total of five categories were extracted: searching for measures tailored to students in a short period of time during training; resistance to individualized responses that significantly differ from traditional Japanese collectivist education; conflict over support being perceived as favoring a particular student; hesitation to identify students’ limits; and barriers in the process of supporting difficulties due to the nature of learning disabilities. Practical training instructors experience difficulties and hesitation when teaching students with potential learning disabilities. The practical training instructors need support and educational opportunities as well as students who need help. To overcome these difficulties, university educational staff, as well as students and families, must be educated on the existence and value of support tailored to the characteristics of an individual’s learning disability.

## 1. Introduction

After being adopted by the United Nations General Assembly, support education for students with learning disabilities was introduced in Japan based on the idea of inclusive education. In 2016, support for university students with disabilities improved as a reasonable accommodation [1]. The number of students diagnosed with learning disabilities is increasing annually, including in nursing and healthcare universities [2,3]. Generally, a learning disability is defined as a condition where a learner faces various difficulties due to being unable to acquire one or more specific abilities, such as listening, speaking, reading, writing, calculating, or reasoning [4]. Prior studies have reported that learning difficulties are often related to on-the-job training [5,6,7]. In addition, a survey of new graduate nurses in Japan found that 66 of them were identified as needing special educational support in 128 medical facilities [8].

In this study, a potential learning disability was defined as a condition in which a lack of integrated ability and communication difficulties were discovered for the first time during clinical practice, and no medical diagnosis was made, despite the student experiencing no problem in learning through classroom lectures.

In Japanese nursing education, clinical training helps to acquire practical ability. This is achieved by integrating, deepening, and verifying the knowledge, skills, and attitudes of specialized subjects based on the knowledge of liberal arts and basic subjects that students have learned in their bachelor’s degrees [9]. Clinical nursing training requires the basic ability to integrate and apply knowledge. Due to learning disability characteristics, difficulties that are difficult to detect at the stage of lectures and exercises in the classroom become apparent in clinical training, for example, care prioritization; nursing record entry by the submission deadline; good communication with patients, other students, and clinical practice instructors; explaining and recording assessments; calculating the amount of drip infusion; flexible response; and the maintenance of mental health [10,11,12]. These difficulties may prevent individuals from achieving the essential goals of clinical practice training in nursing.

According to previous research, to meet the learning needs of each student with a learning disability, teachers need to understand them in terms of the learning situation they experience, rather than simply understanding the disability by name [13]. Students with learning disabilities have called on nursing schools to develop adaptive pathways to become good nurses [14]. If teachers work hard, nursing students with disabilities can become competent nurses, suggesting the need for cross-professional collaboration [15]. In other words, practical training instructors are required to make efforts so that students can achieve essential goals without interfering with patient care and hospital routines.

In Japan, specific support methods for nursing students with learning disabilities are emerging [16]. However, the topic has still not received sufficient research attention, and there are limited reports on the learning support methods that suit the characteristics of nursing students with learning difficulties. For students with potential learning disabilities, practical training instructors work very hard every day. Clinical instructors should also be aided in their support of students with learning disabilities. Therefore, this study aimed to identify the main difficulties experienced by practical training educators when supporting students with learning disabilities.

## 2. Materials and Methods

### 2.1. Design

A qualitative descriptive design was used with online focus group interviews. This was a non-interventional cross-sectional study.

### 2.2. Participants

Nurses with a Bachelor of Nursing degree and having more than five years of clinical teaching experience were recruited through our website. We notified them on the university website, informing the audience of the purpose and method of this research and requesting them to disseminate this information in a snowball manner. The inclusion criteria for participants were to have (1) a bachelor’s degree in nursing, (2) practical training experience in nursing, and (3) experience or interest in supporting students with learning disabilities. The participants were nine Japanese nursing professionals (eight women and one man). Their occupations were health nurses, midwives, registered nurses, school nurses, and university teaching staff. Participants had an average of 15.4 (8–21) years of work experience. Written informed consent was obtained from all the participants. The practicum training educator was a hospital nurse who was in charge of clinical education in addition to being a university teacher.

### 2.3. Data Collection

Data were collected in March 2021. Three interactive focus group interviews were conducted with three participants each. They were scheduled in groups of three at their convenience; thus, three groups were formed. Following an interview guide, we interviewed the participants for approximately 90 min, using Microsoft Teams instead of face-to-face interviews to avoid COVID-19 infection (Table 1). The researchers facilitated the interview using the semi-structured interview guide and recorded each focus group interview with the prior permission of the participants. The three questions were (i) What do you think about the difficulties associated with working with nursing students who may have learning disabilities? (ii) What do you think about learning support in clinical training for nursing students who may have learning disabilities? (iii) What do you think about support for the career selection of nursing students who may have learning disabilities? The process of developing the three question items is outlined as follows. Question (i) was set as an introductory question that was non-directive, easy for participants to answer, and easy for researchers to analyze later. This was because it was expected that the participants would be able to express their thoughts and real words rooted in their specific experiences, and that the relationship between the information and the group members would be clarified through this question. Question (ii) was set as an essential non-directive question to clarify the purpose of this research. While discussing question (i), the participants tended to add their ideas to the opinions of other members to come up with new ideas and make remarks [17]. Therefore, we set it as the peak of the interactive focus group interview, as it was possible to discuss current issues and future measures. Question (iii) was set up to supplement question (ii) because practical training educators are role models for students and their advice often influences future career choices. Before the interview started, the participants introduced themselves as an ice-breaker, and at the end of the interview, they asked candidly about their impressions of participating. Participants could review a transcript with their personal information blinded the day after the interview.

### 2.4. Data Analysis

An accurate transcript was created from the records. The notation has been unified so as not to spoil the contents. A qualitative descriptive analysis was performed after and personally identifiable information was removed. We based this analysis on Mayring’s (1983) qualitative content analysis [18]. The first step in the analysis was to confirm the data and select statements from the transcripts that were relevant to our research objectives. As the second step, we analyzed whether the data collection situation was appropriate for the establishment situation of the data. Third, we classified the materials in terms of form and examined whether the recorded data and verbatim records were correctly transcribed, and whether personal information was deleted. The verbatim records were reviewed by participants. As the fourth step, we set the direction of the analysis, and grasped the “difficulty for practical training educators” in terms of context [18]. Through these processes, the researchers paraphrased individual contextual units by summarizing content analysis. In addition, subcategories were generated by researchers and named based on the similarity of semantic content. The three researchers first held a meeting to reach a consensus on contextually understanding the “difficulties of practicum educators.” In this study, each of the three groups was examined and the work was repeated until it was constant. In the data obtained from three groups, typical, empirical, extreme, diverse, and easy-to-imaginable narratives were scattered across group [17]. Therefore, rather than comparing responses across groups, the categories were generated by combining the data from the three groups. We followed the consolidated criteria for reporting qualitative research (COREQ [19]). Interview recordings were analyzed using NVivo software ver.1.6.

### 2.5. Ethical Considerations

This study was approved by the Ethics Committee of Hiroshima University (approval no. E-1972). Data collection was designed to ensure confidentiality and anonymity; participants provided informed consent before taking part in the study. In addition, they were informed before the interview that their cooperation was voluntary, would result in no disadvantages, and could be interrupted/withdrawn at any time. Descriptions of identifiable individuals or regions were anonymized prior to the analysis.

## 3. Results

A total of 5 categories and 17 subcategories were extracted from the interview data (Table 2). 

### 3.1. Category 1: Searching for Measures Tailored to Students in a Short Period during Training

Clinical nursing instructors and university teaching staff sought methods that would suit students with potential learning disabilities in clinical nursing practice, where their areas of expertise rapidly change in a short period. The typical narratives in each subcategory are as follows. The numbers in parentheses indicate the transcript page numbers of the groups from which the typical narratives were extracted.

#### 3.1.1. Subcategory 1: Conflict over Lowering Goals as a Reasonable Accommodation

This subcategory contained the following narratives: “As nursing education, it is personally doubtful that goal setting is individually offered in practice. There are conflicts and doubts regarding whether this is really okay. However, I believe that I must obey. I have no choice but to abandon it” (A-P2); “… I often wonder if a reasonable accommodation is really good for their future” (B-P10).

#### 3.1.2. Subcategory 2: Incompatibility with Goals to Be Achieved in the Short Term

This subcategory contained the following narratives: “It would be very difficult [for students with learning disabilities] to relate to and understand the textbook descriptions and clinical practices in a short training period” (A-P3); “… It is difficult to determine the permissible range of students who may have learning disabilities during clinical training. I am worried about how to come to terms with what I want to learn and the goals to be achieved in a short training period” (B-P3).

#### 3.1.3. Subcategory 3: Barriers to Prior Information Sharing about the Students to Consider

This subcategory contained the following narratives: “… Ah! If you told me more [clinical training] before, there was a way [suitable for the student] …” (A-P3); “If we know the difficulty or disability in advance, we can coordinate with the clinical training destination, so we think that students with learning disabilities can manage the training without any trouble” (B-P6).

#### 3.1.4. Subcategory 4: Unclear Specific Support Measures Suitable for the Students

This subcategory contained the following narratives: “Rather than the information that this student has a learning disability, I would like to know specific coping methods and support methods tailored to their characteristics” (A-P2); “After evaluating the training, there was a tendency for support to be over, so we had to learn how to support students with learning disabilities” (B-P12).

#### 3.1.5. Subcategory 5: Unfamiliar with Learning Disabilities and Prejudiced

This subcategory contained the following narratives: “Some leaders responded negatively [because it was different from their own experience] when they did not get the expected response from nursing students with learning disabilities because they were unfamiliar with learning disabilities” (A-P1); “… When we were students, we did not study learning disabilities, and we had prejudices against them. Therefore, it was not possible to connect to appropriate support tailored to them” (A-P3); “I didn’t have the basic knowledge, so I was surprised that there were students with learning disabilities who had passed the university entrance examination” (C-P1).

#### 3.1.6. Subcategory 6: Feelings of Experience-Oriented Veteran Clinical Practice Leaders

This subcategory contained the following narratives: “The teaching methods of senior nurses and midwives and veteran clinical training instructors are experience-oriented, and the education they have learned is reflected in their guidance of juniors. It is often quite emotional, and I do not think it is good. [everyone agrees and nods].” (A-P5); “At university, it is easy for faculty members to share reasonable accommodations with students. However, I do not know how to explain reasonable accommodation to the clinical training instructor. There are differences in the degrees of the understanding of learning disabilities among clinical training instructors, depending on the ward and facility” (C-P6).

### 3.2. Category 2: Resistance to Individualized Responses That Significantly Differ from Traditional Japanese Collectivist Education

In this category, various types of resistance were identified in the Japanese collectivist education activity of clinical nursing training, where university education staff and clinical training instructors were forced to individually respond to the characteristics of nursing students with learning disabilities.

#### 3.2.1. Subcategory 1: Learning Disabilities That Tend to Be Hidden by Students Themselves

This subcategory contained the following narratives: “… It is difficult to understand without complaints from students with learning disabilities. It was only noticed after a considerable delay in learning clinical practice” (A-P4);. “... I really agree that an environment in which I can say things that trouble me is very important. The difficulty in working with nursing students with learning disabilities is that they have difficulty seeing their own disabilities and difficulties” (B-P1); “I don’t know how to talk so that the student doesn’t get hurt. … It is very difficult for students with learning disabilities to deal with themselves” (C-P5).

#### 3.2.2. Subcategory 2: Difficulty Distinguishing between Reading and Writing or Reasoning

This subcategory contained the following narratives: “Every year, there were some nursing students who were unable to write nursing records when guiding clinical nursing training. It is difficult to distinguish whether the student is experiencing difficulty writing or cannot reason” (C-P1); “… Certainly, there are students who cannot write nursing records. If university education staff can take the time to listen individually and slowly, they can better understand what [students with learning disabilities] are thinking. If they think, they can urge them to transcribe the idea. But they are so silent… so I do not really know what to do” (C-P2).

#### 3.2.3. Subcategory 3: Resistance to Labeling What May Be a Learning Disability

This subcategory contained the following narratives: “What used to be recognized as a strange student is now labeled as a learning disability. … I think it’s our role to help them respond well, but there are some conflicts” (B-P2); “Since there are no students in Japan who self-report that they have a learning disability, I am confused as to whether it is okay to have prejudice that I may have a learning disability as a teacher” (C-P1).

#### 3.2.4. Subcategory 4: No Understanding of Reasonable Accommodation

This subcategory contained the following narratives: “Even in clinical nursing practice, strict bosses often attack, and all nursing staff are atrophied. … It is important to build a trusting relationship. Relationships are very important while we can talk to each other with peace of mind” (B-P1); “If an individual is diagnosed with a learning disability and it is disclosed, this can be explained to the members of the training group. However, from the viewpoint of personal information protection, it is difficult to balance whether it should be disclosed” (C-P3); “It is difficult to reach a consensus for all because the understanding of reasonable accommodation varies among university education staff” (C-P4).

### 3.3. Category 3: Conflict over Support Being Perceived as Favoring a Particular Student

In this category, we identified the conflict that caring for and supporting the characteristics of nursing students with learning disabilities can adversely affect relationships between group members.

#### 3.3.1. Subcategory 1: Disregarding Learning Disabilities as a Mere Personal Feature

This subcategory contained the following narratives: “… When involved in clinical training without any prior information, [a potential learning disability] is merely considered an individuality of an unmotivated nursing student” (A-P3); “It is only captured by the image that the student’s difficulty does not change. I think that it is a big difficulty that tends to be biased, such as a difficulty only for the students themselves” (B-P2).

#### 3.3.2. Subcategory 2: Adverse Effects on Student Relationships and Group Learning

This subcategory contained the following narratives: “If there is information sharing in advance from the university education staff, we can prepare … I think that the atmosphere between the members of the training group would then be more relaxed” (A-P7); “It is very difficult to improve broken relationships with other students in terms of how to work with them. Even if it is a difficulty between students, it is difficult because they cannot leave it to them” (C-P4); “… Leader students of the same group members as students with learning disabilities seems to be very painful” (C-P3).

### 3.4. Category 4: Hesitation to Identify the Limits of Students

In this category, the hesitation of educational staff was extracted from the experience of negative clinical training regarding whether it was good to identify the limits of students with learning disabilities.

#### Subcategory 1: Hesitation to Identify the Limits from the Clinical Training Experience

This subcategory contained the following narratives: “Clinical nursing training is an opportunity to know yourself by looking at what you are not good at, whether it suits you or not” (A-P8); “From the perspective of educational staff and instructors, this student is worried about working in the field of life or nursing care” (B-P11); “As a teacher in the role of a tutor, it is difficult to judge whether it is okay to proceed on my own and consult with a specialized institution” (C-P5).

### 3.5. Category 5: Barriers in the Process of Supporting Difficulties due to the Nature of Learning Disabilities

In this category, university education staff identified various barriers to supporting the unique difficulties of students with learning disabilities.

#### 3.5.1. Subcategory 1: Unconscious Negative Feedback

This subcategory contained the following narratives: “It is very difficult for the group members to notice and point out what is always pointed out by the teaching staff without being noticed by students who may have learning disabilities” (A-P7); “When the patient and student had a dialogue, the student forgot the request, and the patient complained, which became a big problem. … Even if the training instructor pointed it out, the student interrupted them and was not convinced” (B-P7).

#### 3.5.2. Subcategory 2: Lack of Solutions and Staffing Tailored to Characteristics of Students with Learning Disabilities

This subcategory contained the following narratives: “For example, I wish I had prepared to write in red and blue on the whiteboard, emphasize slowly, and shorten one sentence” (A-P4); “Educational and clinical settings are very busy. To be honest, it is difficult to change politely to one student who needs support” (B-P5); “Personally, when creating handouts for students, I try to make them easy to read and create materials with clear colors and contrasts as a universal design” (C-P7).

#### 3.5.3. Subcategory 3: Efforts to Understand the Characteristics of Students with Learning Disabilities and Build a Relationship of Trust

This subcategory contained the following narratives: “… They may not know if it is their first training, but before the training, they will talk about what they can say whenever they have difficulty and the importance of noticing when they get a sign that they are not good at something while working in various areas. I wish I had an announcement” (A-P7); “Students do not seek support from [being given] information alone, so the university education staff who knew them best often went with them for counseling…” (B-P4); “University education staff will have to listen to the background of the students and get involved. Students with potential learning disabilities tend to be scolded by others from an early age because they are not good at learning. In addition, even if you enter a university, you may be depressed by the negative experience during the training, which may lead to secondary disabilities. We have to consider why the students think this way rather than scolding them bluntly” (C-P5).

#### 3.5.4. Subcategory 4: Not Being Able to Cooperate

This subcategory contained the following narratives: “If an expert intervened with a student with a learning disability during the training and asked them to listen to difficult or troubling things, the instructor could have solved the problem [actually, it was not solved]” (A-P4); “There are places where the university education staff alone cannot afford to be fully involved” (B-P6); “It is not just a tutor; as a university organization, we must support students who may have learning disabilities in cooperating with their parents and psychologists. I thought this was part of faculty development” (C-P9).

## 4. Discussion

This study clarified the difficulties faced by practical training instructors who teach students with potential learning disabilities. The participants were veteran nursing professionals with experience in clinical training. They leveraged a wide range of occupational experiences to provide highly useful narratives about difficult episodes, which revealed the specific burdens of practical training instructors.

The participants used their personal experiences to determine suitable instructional methods for students with potential learning disabilities during short-term nursing training. Even if a student is tutored for an extended period of time in the hope that he or she will grow, if the student feels uncomfortable with being restrained, it may be misconstrued as harassment [20]. It is suggested that practical training leaders should be trained in how to assist with learning disabilities [21]. Moreover, it is necessary to create organizations and rules that facilitate reasonable accommodation. A reasonable accommodation is provided by a university per the condition and characteristics of each student and is diverse and highly individual [1]. Therefore, nursing universities must establish a system that can provide a comprehensive education in practical training using the ingenuity of each educational staff member with clear rules to facilitate this.

Participants were reluctant to focus solely on students with potential learning disabilities because this approach differs from the traditional Japanese collectivist style of education they received. Japanese collectivist education fosters individual independence with a strong sense of solidarity with the group and promotes group unity and discipline based on the concept of “one for all, all for one” [22,23,24]. Presumably, this influenced former elementary, junior high, and high school education as a thinking base for the educational staff. In contrast, supporting various learning methods for students with learning disabilities requires understanding their individual needs [13]. Practical training instructors felt resistance possibly due to the excessive gap between the education they received as students and the unfamiliar reality of supporting students with learning disabilities as educators. Hence, understanding the characteristics of students with learning disabilities is critical so that practical training instructors can dispel the notion of traditional collectivist education and accept modern individualized education [25,26]. Universities should provide training to support practical training instructors.

If a correct understanding of the surroundings is not obtained, giving special consideration to a particular student may be perceived as favoring them. Practical training instructors familiar with the principles of the ethical codes of fairness try to behave fairly toward students [27]. Consequently, it was presumed that the more they give attention to a specific student, the more likely it is that practical training instructors will experience conflicts. Moreover, the social stigma of people with learning disabilities cannot be eliminated [28]. Thus, practical training instructors need to have a strong value base and adopt RICI (Rights, Inclusion, Choice, Independence) principles [29]. Appropriate support requires that both the students receiving the support and the practical training instructors are free from physical and mental stress. Therefore, it is necessary to form project teams that include the students themselves [16] and that not only support students with learning disabilities but also provide meta-support to practical training instructors and surrounding professional staff.

University education staff were confused by the negative experiences of students with learning disabilities in identifying their limits in clinical training. The ability of nursing students with disabilities to become competent in the future is influenced by their educators [15]; therefore, it may be inappropriate to determine their limits by educators. However, with educators determining the limits, students with higher academic resilience could continue learning and strengthen their social emotions [30], meaning that knowing their limits would allow them to appreciate their strengths and seek appropriate career choices. This conclusion is supported by our finding that students have a good opportunity to confront their weaknesses in clinical nursing practice. There are pros and cons to identifying the limitations of promising adolescent students, and further research is needed.

Altogether, education staff should recognize students with learning disabilities and the people around them as a support team and learn from each other to address difficulties in clinical training in Japanese nursing universities. Teams should be aware of clinical teaching educators’ struggles and be careful not to isolate them for their mental health. Furthermore, as a social organization, the university must be responsible for and proactively engage in educational activities related to learning disabilities to provide reasonable accommodation. Our results should trigger such endeavors.

This study has several limitations. First, all participants were recruited from one university in Hiroshima, Japan, and their opinions may not be broadly applicable or comprehensive. Second, few nursing students in Japan accept and publicize the diagnosis of their learning disabilities. Therefore, participants may have vaguely stated their perceptions of “students with potential learning disabilities” in the focus group interviews. Third, we collected data from online focus group interviews. Future research should test the reliability of our findings by similarly analyzing data collected through face-to-face methods; in the future, we need to increase the number of participants and conduct further research on the differences between online and face-to-face focus group interviews.

## 5. Conclusions

Our study revealed five problems experienced by training instructors and teaching staff involved in clinical training for students with potential learning disabilities at a nursing university. These are as follows: ”Searching for measures tailored to students in a short period during training,” “Resistance to individualized responses that differ significantly from traditional Japanese collectivist education,” “Conflict over support being perceived as favoring a particular student,” “Hesitation to identify the limits of students,” and “Barriers in the process of supporting difficulties due to the nature of learning disabilities.” Clinical training educators were struggling, and there is an evident need to support not only the students but also the educators. To overcome these difficulties, university educational staff, as well as students and families, need to be educated about the existence and value of support tailored to the characteristics of an individual’s learning disability.

## Figures and Tables

**Table 1 healthcare-11-00615-t001:** Online focus group interview guide.

Time (Minutes) *	Implementation Details
Before starting	Check audio and web camera when connecting online, and secure a relaxing environment
4	Introduction: Greetings, explanation of research outline, and confirmation of consent to research cooperation
3	Checking the definition of learning disabilities and how to proceed with online focus group interviews
3	Participant self-introduction
25	Question 1. What do you think about the difficulties involved in working with nursing students who may have learning disabilities?
25	Question 2. What do you think about learning support in clinical training for nursing students who may have learning disabilities?
25	Question 3. What do you think about supporting the career selection of nursing students who may have learning disabilities?
5	Summary and upcoming information

* Total time (minutes): approximately 90 min.

**Table 2 healthcare-11-00615-t002:** Categories and subcategories of the difficulties faced by clinical instructors and university education staff working with nursing students with potential learning disabilities.

Categories	Subcategories
Searching for measures tailored to students in a short period during training	Conflict to lower goals as a reasonable accommodation
	Incompatibility with goals to be achieved in the short term
	Barriers to prior information-sharing about the students to consider
	Unclear specific support measures suitable for the students
	Unfamiliar with learning disabilities and prejudiced
	Feelings of experience-oriented veteran clinical practice leaders
Resistance to individualized responses that differ significantly from traditional Japanese collectivist education	Learning disabilities that tend to be hidden by students themselves
	Difficulty in distinguishing between reading and writing or reasoning
	Resistance to labeling what may be a learning disability
	No understanding of reasonable accommodation
Conflict over support being perceived as favoring a particular student	Disregarding learning disabilities as a mere personal feature
	Adverse effects on student relationships and group learning
Hesitation to identify the limits of students	Hesitation to identify students’ limits from the clinical training experience
Barriers in the process of supporting difficulties due to the nature of learning disabilities	Unconscious negative feedback
	Lack of solutions and staffing tailored to the characteristics of students with learning disabilities
	Efforts to understand the characteristics of students with learning disabilities and build a relationship of trust
	Not being able to cooperate

## Data Availability

The data that support the findings of this study are available from the corresponding author upon reasonable request.

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
