# Peer review of "Distress of Educators Teaching Nursing Students with Potential Learning Disabilities: A Qualitative Analysis"

_healthcare, 2023, doi:10.3390/healthcare11040615_

Round 1

Author Response

Thank you for your very useful comments. Please confirm that it has been corrected as follows.

Point 1: The title states it is about the mental health of educators, but this was never studied. Please edit the title to match the paper.

Response 1: Thank you for your suggestion. We have added and revised the text as follows: “Distress of Educators Teaching Nursing Students with Potential Learning Disabilities: A Qualitative Analysis." (Page 1, line 2-3).

Point 2: The abstract has a purpose. The terminology practical training instructors and university education staff will need to be clariid in the paper since these are not necessarily terms used in other countries. Participants are listed as Japanese nursing university graduates which implies the participants were students. The description of the sample in the abstract and paper is confusing and needs to be corrected.

Response 2: Thank you for your comments. We have added and revised the abstract as follows: “Clinical training at Japanese nursing universities has an increasing need for individualized learning support for students with potential learning disabilities. Despite a high interest in student support, educators' difficulties are neglected. This study clarified the difficulties encountered by practical training instructors in delivering clinical training to nursing students with potential learning disabilities. In this descriptive, qualitative study, online focus group interviews were conducted. Participants were nine Japanese nursing university graduates with over five years of clinical education experience. Five categories were extracted: searching for measures tailored to students in a short period of time during training; resistance to individualized responses that significantly differ from traditional Japanese collectivist education; conflict over support being perceived as favoring a particular student; hesitation to identify students’ limits; and barriers in the process of supporting difficulties due to the nature of learning disabilities. Practical training instructors experience difficulties and hesitation when teaching students with potential learning disabilities. The practical training instructors need support and educational opportunities as well as students who need help. To overcome these difficulties, university educational staff, as well as students and families, must be educated on the existence and value of support tailored to the characteristics of an individual’s learning disability." (Page 1, lines 9-24).

 “Participants” was revised to “practice training instructors” throughout. This was also modified in the abstract, to eliminate any confusion regarding whether participants were instructors or students.

Point 3: The design is stated as qualitative. Usually, researchers will state the qualitative theory that was used since the theory can drive the data collected and how it is analyzed.

Response 3: Thank you for your comments. A qualitative descriptive analysis was performed for hypothesis generation. Descriptions of difficulties are extracted by software.

Point 4: In section 2.2 university graduates are stated. Do you mean faculty? Further down the authors state Japanese students. Were participants faculty or students?

Response 4: Thank you for your comments. We have added and revised the text as follows: “Nurses with a Bachelor of Nursing degree and having more than five years of clinical teaching experience were recruited through our website." (Page 2, line 77-78).

Point 5: The focus group implementation is nicely described and is the content for Table 1.

Response 5: Thank you very much your comments.

Point 6: IRB approval is stated.

Response 6: Thank you very much your comments.

Point 7: The categories and subcategories are listed in a table and briefly described. Can the 19 subcategories be condensed? Some are a challenge to understand.

Response 7: Thank you for your comments. We have added and revised the text as follows: “Five categories and 17 subcategories were extracted from the interview data (Table 2)." (Page 4, line 157-158).

Point 8: In the discussion, what is the difference between university teachers and clinical training instructors? This is a country/culture issue and needs to be clarified/described early in the paper. If there are differences in these two groups, why did you not keep the findings consistent and only use one type of faculty? The sample size is so small that having 2 different types of faculties can affect findings.

Response 8: We thank the reviewer for this insightful suggestion. University teachers and clinical training instructors have the same meaning. We have added and revised the discussion (Pages 8-10, lines 327-394).

Point 9: On page 9, there is a paragraph about barriers to addressing teaching difficulties. I do not believe this is the focus of the paper. Is this for future research? Did this study give you information to focus on one type of teaching issue at a time such as the issue that is most common?

Response 9: We thank the reviewer for this insightful suggestion. This part has been deleted.

Point 10: Limitations are stated but I do not understand line 370.

Response 10: Thank you for your suggestion. We have added and revised the text as follows: “Therefore, participants might have vaguely stated their perceptions of “students with potential learning disabilities” in the focus group interviews. Third, we collected data from online focus group interviews. Future research should test the reliability of our findings by similarly analyzing data collected through face-to-face methods; in the future, we need to increase the number of participants and conduct further research on the differences between online and face-to-face focus group interviews." (Pages 9-10, line 389-394).

Point 11: References appear to be ok.

Response 11: Thank you very much your comments.

Reviewer 2 Report

This is a psychological study on learning disabilities from Japanese nursing college students. I think this topic is very valuable, and the full text has a basic writing paradigm and good writing. But because this is a qualitative study, I think the authors lack some important reports on the analysis process.

The first part is the introduction. Authors should add some descriptions about the current situation in Japan, instead of replacing them with insufficient attention. If we don't pay attention to it, we should also increase the appropriate explanation of the reasons for this situation, which is also the important value and significance of further explaining this study.

The second is materials and methods. I suggest that the authors explain why these three questions are determined. Is there any basis? So that readers can quickly ensure the science of this study. Are there any other questions in the interview?

The third is the result. I think this part is the most important part of the full text, but the author did the worst part. First, there is no theoretical basis for the dimension division of results, and it is hard to guarantee that it is not self-explanatory. Secondly, the analysis process of qualitative research is obviously not so simple. The authors should inform the steps to reach these conclusions, such as structural coding, spindle coding, process coding and so on. Third, in line 146, 3.1.3, the authors should have left out "subcategory". I suggest that the authors should rewrite this part and refer to the reporting process of some mature published qualitative studies.

Finally, the conclusion. We all know the importance and reference of research. However, the authors should report your unique thinking in the research, which can enlighten Japan and even the whole world to pay attention to people with learning disabilities, or what measures can be tried to intervene.

I hope the authors can take your research seriously, and  can get a great manuscript after revision. I look forward to your revised paper.

Author Response

Thank you for your very useful comments. Please confirm that the changes now address your concerns.

Point 1: This is a psychological study on learning disabilities from Japanese nursing college students. I think this topic is very valuable, and the full text has a basic writing paradigm and good writing. But because this is a qualitative study, I think the authors lack some important reports on the analysis process.

Response 1:. Thank you for your helpful suggestion. The extraction process is described in the text. I also added an example of the part where the description about "difficulty" was extracted. We have added and revised the text as follows: “A qualitative descriptive analysis was performed after an accurate transcript was created from the records and personally identifiable information was removed. The interview transcripts were analyzed using NVivo software ver.1.6. NVivo was used for managing verbatim transcripts, coding, search history, history of analyzed results, data merging, and grouping [18]. We based this analysis on Mayring's(1983) qualitative content analysis [19]. The first step in the analysis was to confirm the data and selected statements from the transcripts that were relevant to our research objectives [19]. As the second step, we analyzed whether the data collection situation was appropriate for the establishment situation of the data [19]. Third, we classified the materials in terms of form and examined whether the recorded data and verbatim records were correctly transcribed, and whether personal information was deleted [19]. As the fourth step, we set the direction of the analysis, and grasped the "difficulty for practical training educators" in terms of context [19]. Through these processes, individual contextual units were paraphrased by summarizing content analysis, and subcategories and categories were generated and named based on the similarity of semantic content.

 In the data obtained from three groups, typical, empirical, extreme, diverse, and easy-to-imaginable narratives were scattered across group [17]. Therefore, rather than comparing responses across groups, the data from the three groups were combined for analysis.” (Pages 3-4, lines 120–138).

[17] Name, T. Group Interview Method in Human Service III/Dissertation preparation Development of qualitative research methods based on scientific evidence. Ishiyaku Publishing: Tokyo, Japan, 2010; p.13.

[18] Díaz Hernández, E.L.; Brito Brito, P.R.; García Hernández, A.M. Happiness and mental disorders. Healthcare (Basel). 2022, 10, 1781. DOI:10.3390/healthcare10091781.

[19] Flick, U. Introduction to Qualitative Research. Coding and Categories [New Edition]; Shunjunsha Publishing Company: Tokyo, Japan, 2021; pp. 393-400.

Point 2: The first part is the introduction. Authors should add some descriptions about the current situation in Japan, instead of replacing them with insufficient attention. If we don't pay attention to it, we should also increase the appropriate explanation of the reasons for this situation, which is also the important value and significance of further explaining this study.

Response 2: Thank you for your input. The clinical leader's situation has been added .The text has been revised as follows:” In other words, practical training instructors are required to make efforts so that students can achieve essential goals without interfering with patient care and hospital routines." (Page 2, lines 62-63).

 “For students with potential learning disabilities, practical training instructors work very hard every day. Clinical instructors should also be aided in their support for students with learning disabilities. Therefore, this study aimed to identify the main difficulties experienced by practical training educators when supporting students with learning disabilities.”(Page 2, lines 67-71).

Point 3: The second is materials and methods. I suggest that the authors explain why these three questions are determined. Is there any basis? So that readers can quickly ensure the science of this study. Are there any other questions in the interview?

Response 3: Thank you for your helpful suggestion. This study aims to identify the main difficulties experienced by practical training educators when supporting students with learning disabilities. Therefore, `”Question (i) What do you think about the difficulties associated with working with nursing students who may have learning disabilities?” was set as an introductory question that was non-directive, easy for participants to answer, and easy for researchers to analyze later. This was because it was expected that the participants would be able to express their thoughts and real words rooted in their specific experiences and that the relationship between the information and the group members would be clarified through this question. “Question (ii) What do you think about learning support in clinical training for nursing students who may have learning disabilities?” was set as an essential non-directive question to clarify the purpose of this research. While discussing question (i), the participants tended to add their ideas to the opinions of other members to come up with new ideas and make remarks [17]. Therefore, we set it as the peak of an interactive focus group interview, as it was possible to discuss current issues and future measures. “Question (iii) What do you think about support for the career selection of nursing students who may have learning disabilities?” was set up to supplement question (ii) because practical training educators are role models for students and their advice often influences future career choices. Before the interview started, the participants introduced themselves as an ice breaker, and at the end of the interview, they asked candidly about their impressions of participating. (Page 3, lines 102–116).

[17] Name, T. Group Interview Method in Human Service III/Dissertation preparation Development of qualitative research methods based on scientific evidence. Ishiyaku Publishing: Tokyo, Japan, 2010; p.13.

Point 4: The third is the result. I think this part is the most important part of the full text, but the author did the worst part. First, there is no theoretical basis for the dimension division of results, and it is hard to guarantee that it is not self-explanatory. Secondly, the analysis process of qualitative research is obviously not so simple. The authors should inform the steps to reach these conclusions, such as structural coding, spindle coding, process coding and so on. Third, in line 146, 3.1.3, the authors should have left out "subcategory". I suggest that the authors should rewrite this part and refer to the reporting process of some mature published qualitative studies.

Response 4: Thank you for your helpful suggestion. We have added and revised the text as follows: “The interview transcripts were analyzed using NVivo software ver.1.6. NVivo was used for managing verbatim transcripts, coding, search history, history of analyzed results, data merging, and grouping [18]. We based this analysis on Mayring's(1983) qualitative content analysis [19]. The first step in the analysis was to confirm the data and selected statements from the transcripts that were relevant to our research objectives [19]. As the second step, we analyzed whether the data collection situation was appropriate for the establishment situation of the data [19]. Third, we classified the materials in terms of form and examined whether the recorded data and verbatim records were correctly transcribed, and whether personal information was deleted [19]. As the fourth step, we set the direction of the analysis, and grasped the "difficulty for practical training educators" in terms of context [19]. Through these processes, individual contextual units were paraphrased by summarizing content analysis, and subcategories and categories were generated and named based on the similarity of semantic content.” (Pages 3-4, lines 121–134).

[18] Díaz Hernández, E.L.; Brito Brito, P.R.; García Hernández, A.M. Happiness and mental disorders. Healthcare (Basel). 2022, 10, 1781. DOI:10.3390/healthcare10091781.

[19] Flick, U. Introduction to Qualitative Research. Coding and Categories [New Edition]; Shunjunsha Publishing Company: Tokyo, Japan, 2021; pp. 393-400.

Point 5: Finally, the conclusion. We all know the importance and reference of research. However, the authors should report your unique thinking in the research, which can enlighten Japan and even the whole world to pay attention to people with learning disabilities, or what measures can be tried to intervene

Response 5: Thank you for your helpful suggestion. The point is not only to pay attention to people with learning disabilities but also to those who support them. We have added and revised the text as follows: “Our study revealed five problems experienced by training instructors and teaching staff involved in clinical training for students with potential learning disabilities at a nursing university. There are: ”Searching for measures tailored to students in a short period during training," "Resistance to individualized responses that differ significantly from traditional Japanese collectivist education,” “Conflict over support being perceived as favoring a particular student ," "Hesitation to identify the limits of students," and "Barriers in the process of supporting difficulties due to the nature of learning disabilities." Clinical training educators were struggling, and there is an evident need to support not only the students but also the educators. To overcome these difficulties, university educational staff, as well as students and families, need to be educated about the existence and value of support tailored to the characteristics of an individual’s learning disability. "(Page 10, lines 396–406). 

Point 6: I hope the authors can take your research seriously, and  can get a great manuscript after revision. I look forward to your revised paper.

Response 6: Thank you very much. We have done our best to address your pertinent questions.

Reviewer 3 Report

Lines 34-39: please revise the definition for learning disabilities: clarity and conciseness

Line 60-61: clarify, is the article discussing learning disabilities or 'developmental' disabilities? These are two different disabilities.

Line 69: The authors state an allotment of 90 minutes for each focus group discussion, with 25 minutes for each research question. Did the focus group interviews last the full 90 minutes? Did you need more time or less? Were the focus group sessions interactive?

Lines 72-83: Clarify who your participants are. Are you using 3 focus groups, comparing responses between groups (empirical data) or qualitative responses (thematic development)? I did not see any empirical data analyses

Line 289: general discussion-great intro. Kudos.

Somehow, there needs to be a section concerning identification of individuals with learning disabilities, or else how are you determining classroom and clinical instruction?  (nursing students and graduate license registered nurses). This is a delicate area in approaching persons with any kind of disability, let alone one involving learning.

General discussion: Is the focus of the article instructors (university teachers, clinical training instructors) or those identified participants with learning disabilities? This part should be clarified further.

More current references could be included to support the current status of classroom and clinical instruction for those with identified learning disabilities.

Author Response

Thank you for your insightful comments. Please confirm that the appropriate changes have been made.

Point 1: Lines 34-39: please revise the definition for learning disabilities: clarity and conciseness.

Response 1: Thank you for your helpful suggestion. In this study, we briefly described the definition of learning disabilities provided by the Japanese Ministry of Education, Culture, Sports, Science and Technology. We have added and revised the text as follows: “Generally, a learning disability is defined as a condition where a learner faces various difficulties due to being unable to acquire one or more specific abilities, such as listening, speaking, reading, writing, calculating, or reasoning [4]. Prior studies have reported that learning difficulties are often related to on-the-job training [5-7]. In addition, a survey of new graduate nurses in Japan found that 66 of them were identified as needing special educational support in 128 medical facilities [8].

 In this study, a potential learning disability was defined as a condition in which a lack of integrated ability and communication difficulties were discovered for the first time during clinical practice, and no medical diagnosis was made, despite the student experiencing no problem in learning through classroom lectures."(Page 1, line 34-43).

[4] Ministry of Education, Culture, Sports, Science, and Technology. Definition of learning disability. Available online: https://www.mext.go.jp/a_menu/shotou/tokubetu/mext_00808.html (accessed on 01 Dec 2021).

[5] Ikematsu, Y. National survey on Nursese/Nursing students with suspected developmental disorder. Annual Report on Research Results. Available online: https://kaken.nii.ac.jp/ja/file/KAKENHI-PROJECT-21659496/21659496seika.pdf(accessed on 07 Febr 2023), 2011.

[6] Ikematsu, Y.; Mizutani, M.; Tozaka, H.; Mori, S.; Egawa, K.; Endo, M.; Yokouchi, M. Nursing students with special educational needs in Japan. Nurse Educ. Pract. 2014, 14, 674–679. DOI:10.1016/j.nepr.2014.08.007.

[7] Yamashita, T.; Tokumoto, H. State of learning difficulties during clinical training among nursing students who have or are suspected to have developmental disorders. Bulletin of School of Nursing; Saitama Medical University 2016, 9, 11–17.

[8] Ikematsu, Y.; Egawa, K.; Endo, M. Prevalence and retention status of new graduate nurses with special support needs in Japan. Nurse Educ. Pract. 2019, 36, 28–33. DOI:10.1016/j.nepr.2019.02.007.

Point 2: Line 60-61: clarify, is the article discussing learning disabilities or 'developmental' disabilities? These are two different disabilities.

Response 2: Thank you for your helpful suggestion. Changes have been made to avoid confusion. We have added and revised the text as follows: “In Japan, specific support methods for nursing students with learning disabilities are emerging [16]. However, the topic has still not received sufficient research attention, and there are limited reports on the learning support methods that suit the characteristics of nursing students with learning difficulties. For students with potential learning disabilities, practical training instructors work very hard every day. Clinical instructors should also be aided in their support for students with learning disabilities. Therefore, this study aimed to identify the main difficulties experienced by practical training educators when supporting students with learning disabilities."(Page 2, lines 64-71).

[16] Kitagawa, A. Basics and Practices of Support for Nursing Staff and Nursing Students with Developmental Disabilities; Medical View, Tokyo, Japan, 2020.

Point 3: Line 69: The authors state an allotment of 90 minutes for each focus group discussion, with 25 minutes for each research question. Did the focus group interviews last the full 90 minutes? Did you need more time or less? Were the focus group sessions interactive?

Response 3: Thank you for your helpful suggestion. The focus group interview lasted around 80–93 minutes. We have added and revised the text as follows: “*Total time (minutes): approximately 90 min."(Page 3, line 118).

 The focus group interviews were interactive, so we have mentioned interactive focus group interview. We have revised the text as follows: “Data were collected in March 2021. Three interactive focus group interviews were conducted with three participants each."(Page 2, lines 90-91).

 “Therefore, we set it as the peak of the interactive focus group interview, as it was possible to discuss current issues and future measures."(Page 3, lines 109–111).

Point 4: Lines 72-83: Clarify who your participants are. Are you using 3 focus groups, comparing responses between groups (empirical data) or qualitative responses (thematic development)? I did not see any empirical data analyses.

Response 4: Thank you for your helpful comments. For all three groups, the extracted data were interspersed with typical, empirical, extreme, diverse, and imaginable narratives. Therefore, rather than comparing responses across groups, the data from the three groups were combined for analysis. We have added and revised the text as follows: “In the data obtained from three groups, typical, empirical, extreme, diverse, and easy-to-imaginable narratives were scattered across groups [17]. Therefore, rather than comparing responses across groups, the data from the three groups were combined for analysis." (Page 4, lines 135–138).

[17] Anme, T. Group Interview Method in Human Service III/Dissertation preparation Development of qualitative research methods based on scientific evidence; Ishiyaku Publishing: Tokyo, Japan, 2010; p. 13.

Point 5: Line 289: general discussion-great intro. Kudos.

Response 5: We thank the reviewer for the thorough review of our manuscript.

Point 6: Somehow, there needs to be a section concerning identification of individuals with learning disabilities, or else how are you determining classroom and clinical instruction?  (nursing students and graduate license registered nurses). This is a delicate area in approaching persons with any kind of disability, let alone one involving learning.

Response 6: Thank you for your helpful comments. The focus of this study is on clinical leaders. I've removed the student support book specific measures so that the focus is clear.

Point 7: General discussion: Is the focus of the article instructors (university teachers, clinical training instructors) or those identified participants with learning disabilities? This part should be clarified further.

Response 7: Thank you for your helpful comments. Unclear wording has been fixed in consideration of your suggestion.

Point 8: More current references could be included to support the current status of classroom and clinical instruction for those with identified learning disabilities

Response 8: Thank you for your helpful comments. We have revised the text as follows: “Prior studies have reported that learning difficulties are often related to on-the-job training [5-7]. In addition, a survey of new graduate nurses in Japan found that 66 of them were identified as needing special educational support in 128 medical facilities [8]."(Page 1, lines 36-39).

 “Even if a student is tutored for an extended period of time in the hope that he or she will grow, if the student feels uncomfortable with being restrained, it may be misconstrued as harassment [21].” (Page 8, 332-334).

[5] Ikematsu, Y. National survey on Nursese/Nursing students with suspected developmental disorder. Annual Report on Research Results. Available online: https://kaken.nii.ac.jp/ja/file/KAKENHI-PROJECT-21659496/21659496seika.pdf (accessed on 07 Febr 2023), 2011.

[6] Ikematsu, Y.; Mizutani, M.; Tozaka, H.; Mori, S.; Egawa, K.; Endo, M.; Yokouchi, M. Nursing students with special educational needs in Japan. Nurse Educ. Pract. 2014, 14, 674–679. DOI:10.1016/j.nepr.2014.08.007.

[7] Yamashita, T.; Tokumoto, H. State of learning difficulties during clinical training among nursing students who have or are suspected to have developmental disorders. Bulletin of School of Nursing; Saitama Medical University 2016, 9, 11–17.

[8] Ikematsu, Y.; Egawa, K.; Endo, M. Prevalence and retention status of new graduate nurses with special support needs in Japan. Nurse Educ. Pract. 2019, 36, 28–33. DOI:10.1016/j.nepr.2019.02.007.

[21] Yoshimura, K. Harassment response in nursing educational institutions case 1: Academic harassment by academic advisor. Nursing Education 2021, 62, 1024–1035.

Round 2

Reviewer 2 Report

I think the authors still haven't made a good revision to the previous questions. For qualitative research, strong theoretical support is very necessary, and obviously the authors have not done it. Although the authors state their basis, the elements extracted from the whole study according to Nvivo are only the authors' personal opinions? Nvivo is only a data analysis tool, not a theoretical basis. Have the extracted factors been proofread for deviation? Tested by others? Obviously, none of these authors have done it. The result is that the authors think that these dimensions can only be self-explanatory, and the results are difficult to verify, which also shows the unreliability of the results. I hope that the authors will not replace it with one sentence, but should make a more scientific explanation for the analysis of the results.

Author Response

Point 1: I think the authors still haven't made a good revision to the previous questions. For qualitative research, strong theoretical support is very necessary, and obviously the authors have not done it. Although the authors state their basis, the elements extracted from the whole study according to Nvivo are only the authors' personal opinions? Nvivo is only a data analysis tool, not a theoretical basis. Have the extracted factors been proofread for deviation? Tested by others? Obviously, none of these authors have done it. The result is that the authors think that these dimensions can only be self-explanatory, and the results are difficult to verify, which also shows the unreliability of the results. I hope that the authors will not replace it with one sentence, but should make a more scientific explanation for the analysis of the results

Response 1:. Thank you for your helpful suggestion. We have added and revised the text as follows: “An accurate transcript was created from the records. The notation has been unified so as not to spoil the contents. A qualitative descriptive analysis was performed after and personally identifiable information was removed. We based this analysis on Mayring's(1983) qualitative content analysis [18]. The first step in the analysis was to confirm the data and selected statements from the transcripts that were relevant to our research objectives. As the second step, we analyzed whether the data collection situation was appropriate for the establishment situation of the data. Third, we classified the materials in terms of form and examined whether the recorded data and verbatim records were correctly transcribed, and whether personal information was deleted . The verbatim records were reviewed by participants. As the fourth step, we set the direction of the analysis, and grasped the "difficulty for practical training educators" in terms of context [18]. Through these processes, the researchers paraphrased individual contextual units by summarizing content analysis. In addition, subcategories were generated by researchers and named based on the similarity of semantic content. The three researchers first held a meeting to reach a consensus on contextually understanding the "difficulties of practicum educators." In this study, each of the three groups was examined and the work was repeated until it was constant. In the data obtained from three groups, typical, empirical, extreme, diverse, and easy-to-imaginable narratives were scattered across group [17]. Therefore, rather than comparing responses across groups, the categories were generated by combining the data from the three groups. We followed the consolidated criteria for reporting qualitative re-search (COREQ [19]). Interview recordings were analyzed using NVivo software ver.1.6.” (Pages 3-4, lines 120–141).

[17] Anme, T. Group Interview Method in Human Service III/Dissertation preparation Development of qualitative research methods based on scientific evidence. Ishiyaku Publishing: Tokyo, Japan, 2010; p.13.

[18] Flick, U. Introduction to Qualitative Research. Coding and Categories [New Edition]; Shunjunsha Publishing Company: Tokyo, Japan, 2021; pp. 393-400.

[19] Tong, A.; Sainsbury, P.; Craig, J. Consolidated criteria for reporting qualitative research (COREQ): A 32-item checklist for interviews and focus groups. Int. J. Qual. Health Care. 2007, 19, 349–357. DOI: 10.1093/intqhc/mzm042.